# Subjective Sleep Quality Versus Objective Accelerometric Measures of Sleep and Systemic Concentrations of Sleep-Related Hormones as Objective Biomarkers in Fibromyalgia Patients

**DOI:** 10.3390/biomedicines11071980

**Published:** 2023-07-13

**Authors:** María Dolores Hinchado, Eduardo Otero, Isabel Gálvez, Leticia Martín-Cordero, María del Carmen Navarro, Eduardo Ortega

**Affiliations:** 1Immunophysiology Research Group, Instituto Universitario de Investigación Biosanitaria de Extremadura (INUBE), Av. de Elvas s/n, 06080 Badajoz, Spain; mhinsan@unex.es (M.D.H.); igalvez@unex.es (I.G.); leticiamartin@unex.es (L.M.-C.); cnavarropz@unex.es (M.d.C.N.); orincon@unex.es (E.O.); 2Immunophysiology Research Group, Physiology Department, Faculty of Sciences, University of Extremadura, 06071 Badajoz, Spain; 3Immunophysiology Research Group, Nursing Department, Faculty of Medicine and Health Sciences, University of Extremadura, 06071 Badajoz, Spain; 4Immunophysiology Research Group, Nursing Department, University Center of Plasencia, University of Extremadura, 10600 Plasencia, Spain

**Keywords:** fibromyalgia, sleep, accelerometry, serotonin, melatonin, catecholamines

## Abstract

Poor quality of sleep leads to an increase in severity of the symptoms associated with fibromyalgia (FM) syndrome and vice versa. The aim of this study was to determine if the poor perceived sleep quality in FM patients could be corroborated by objective physiological determinations. Perceived sleep quality was evaluated (through the Pittsburgh Sleep Quality Index) in 68 FM patients compared to an age-matched reference group of 68 women without FM. Objective sleep quality (measured using accelerometry), and systemic concentrations of sleep-related hormones (catecholamines, oxytocin, serotonin, and melatonin) were evaluated in two representative groups from the reference control group (*n* = 11) and FM patients (*n* = 11). FM patients reported poorer subjective sleep quality compared to the reference group. However, no significant differences were found in accelerometry parameters, except for a delay in getting in and out of bed. In addition, FM patients showed no significant differences in oxytocin concentration and adrenaline/noradrenaline ratio, as well as a lower serotonin/melatonin ratio. Poor perception of sleep quality in FM patients does not correspond to objective determinations. A dysregulation of the stress response could be associated with the delay in their resting circadian rhythm and difficulty falling asleep. This would be the cause that justifies the perceived lack of rest and the fatigue they feel when waking up.

## 1. Introduction

While pain and fatigue are the most frequent symptoms associated with fibromyalgia (FM), most patients with this syndrome also suffer from sleep disturbances and cognitive and mood alterations [1,2]. Although numerous precedents have highlighted the role of sleep in this syndrome [3], the substantial importance of sleep disturbances and sleep quality has only recently been recognised in the aetiology of FM. In fact, the American College of Rheumatology (ACR) proposed a modification to the diagnostic criteria for FM in 2010, which, in addition to removing trigger points, emphasised the assessment of the severity of sleep problems and fatigue [4].

Sleep problems are known to be related to nonspecific pain: pain leads to decreased sleep, and sleep deprivation leads to pain, producing a repetitive circle of decreased sleep and increased pain [5,6]. It has been suggested that two thirds of FM patients have sleep disturbances, but this is not considered an underlying component in its aetiopathogenesis. However, it is accepted that there is a bidirectional relationship in which poor sleep quality leads to increased pain severity and poor cognitive performance in patients with FM [7], clearly constituting a reciprocal relationship [8,9]. Furthermore, this poor sleep quality reported in FM patients, generally assessed only subjectively, leads to worsening symptoms such as depression [10], emotional distress [11,12,13], and difficulties with memory and attention [8,14] that usually develop throughout the course of this disease [15].

Sleep is a complex physiological process related to the preservation of homeostasis and neuroplasticity, regulated globally and locally both by cellular and molecular mechanisms [16]. Most sleep researchers agree with the fact that sleep having a single function is not a realistic view, because sleep is involved in numerous vital physiological functions, such as the development and conservation of energy, modulation of immune responses, cognition, and psychological conditions [17,18], all of which have been reported to be deteriorated in FM syndrome [15,19]. Many neurotransmitters and hormones are involved in sleep regulation [20], including catecholamines (adrenaline, noradrenaline), serotonin [21,22], oxytocin [23,24], and melatonin [25,26].

In this context, as a continuation of our previous study in the present special issue on “Advanced Research on Fibromyalgia” [27], the main objective of the present investigation was to determine if the poor perceived sleep quality in FM patients could be corroborated by objective determinations, such as objective accelerometry tests and ratios of the systemic concentrations of hormones and neuromediators related to good or bad sleep quality, particularly focused on the adrenaline/noradrenaline and serotonin/melatonin ratios.

## 2. Materials and Methods

### 2.1. Participants and Experimental Design

This study was carried out on 68 women diagnosed with FM (FM patient group) belonging to the FM association EXISTIMOS ^®^ (Badajoz, Extremadura, Spain; Extremadura is a reference region in the FM investigation due to the homogeneous population in terms of lifestyle [28,29]). All participants included in the study were within the age range of 40 to 65 years. Sixty-eight women of the same age range were used as a reference group (RG) of “healthy” women, not diagnosed with FM, CFS, any other inflammatory or rheumatic pathology, or any pathology that could potentially affect sleep quality. All volunteers from the FM association who met the inclusion criteria were selected: (a) diagnosis of FM by rheumatologists or internal medicine professionals according to ACR diagnostic criteria [30]; (b) aged between 40 and 65 years; (c) not having a diagnosis of depression; (d) not suffering from multiple chemical sensitivity; and (e) not taking corticosteroids; all of them in accordance with our previous investigations [15,27]. In the first part of the study, participants filled in the Pittsburgh Sleep Quality Index (PSQI) under supervision and with the corresponding instructions in due time and proper course.

In the second part of the study (objective determinations), a representative group from the RG and “FM patient” group were selected (RRG = 11 and RFM = 11, respectively). This reduced number of representative volunteers from each group (all volunteers were in the range of the same anthropometric characteristics and PSQI score than the RG and FM patient groups) allowed us to objectively determine the sleep quality and hormones related to sleep and stress, at the same time with the same devices, and in the same neuroendocrine assays, without interassay variations.

In order to obtain more information and given that pace of life could differ depending on whether it was a weekend or a weekday, the accelerometry study was carried out over a period of seven days. Serum and saliva samples were collected at 08:00 a.m. after the last accelerometry test to determine the systemic concentration of serotonin, oxytocin, adrenaline, noradrenaline, and melatonin using ELISA. It is important to note that on the night of saliva collection, participants were monitored with an accelerometer to ensure that no other sleep disturbances occurred that night.

Written informed consent was also requested from all participants before participating in the study. The research had been previously approved by the Bioethics Committee of the University of Extremadura by the Directives of the Council of Europe and the Declaration of Helsinki (registration number 13/2020). This study was registered with ClinicalTests.gov (identifier: NCT05323838—available on the website).

Table 1 shows the main characteristics of the participants: anthropometric data and PSQI score. At the time of the present study, all of patients were prescribed with different types of anti-inflammatory drugs (e.g., tramadol, acetaminophen, ibuprofen). Prescription drugs related to improving sleep quality were restricted.

### 2.2. Subjective Determination of Sleep Quality: Pittsburgh Sleep Quality Index (PSQI)

The PSQI is one of the most frequently used instruments for subjective sleep assessment. It has appropriate internal consistency, sensitivity, and specificity for the assessment of sleep in primary insomnia [31]. The Spanish version of the PSQI used by Hita-Contreras and co-workers [32] provides a robust instrument with good cross-validity for measuring sleep quality among Spanish patients with FM.

### 2.3. Objective Determination of Sleep Quality: Accelerometry

Objective levels of sleep quality were evaluated following previous studies from our laboratory [27]. Briefly, the Actigraph wGT3X-BT accelerometer was used to measure different objective parameters related to sleep quality: in-bed and out-bed times, latency, efficiency, wake after sleep onset (WASO), number of awakenings, and daily lux average counts. Participants wore the accelerometer on the nondominant wrist for seven consecutive days without interruption, except during showers or any water-related activity, which could disrupt proper functioning. Subsequently, the files generated by the accelerometer were analysed using a specific software called Actilifie 6 (ActiGraph, LLC, Pensacola, FL, USA) using the Cole–Kripke algorithm [33].

### 2.4. Saliva and Blood Samples

Blood samples were collected from fasting subjects at 08:00 a.m. on the same day of actigraphic device collection and placed in collection tubes for serum isolation, where they were kept for 15–20 min at room temperature. The serum was centrifuged at 1800× *g* for 15 min. Serum samples were coded and gradually refrigerated at −20 °C as they were obtained. Finally, the samples were stored at −80 °C until further analysis.

Saliva samples were extracted using a non-invasive method (collection methods: SalivaBio Oral Swab, Salimetrics, Carlsbad, CA, USA) at 08:00 a.m. Participants were required not to ingest any food or drink containing sugars, alcohol and/or caffeine, or tobacco at least 12 h before testing. Volunteers were requested to open the container and remove the sterile swab and place it correctly in the mouth, under the tongue, and were recommended to hold it for at least 2 min to ensure that there were no fluctuations in the sample volume. Immediately following this, the samples were refrigerated at −20 °C and finally stored at −80 °C until further analysis.

### 2.5. Determination of Neuroendocrine Markers

Serum concentrations of oxytocin (CloudClone Corp., Katy, TX, USA), serotonin (Reddot Biotech. Inc., Katy, TX, USA), adrenaline, and noradrenaline (Demeditec Diagnostic GmbH, Kiel, Germany) were measured using commercial ELISA kits. Salivary melatonin concentrations were also measured using commercial ELISA kits (Salimetrics, Carlsbad, CA, USA). Melatonin levels in serum parallel the corresponding variations in saliva, where salivary concentrations are approximately 30% of those found in serum, with a high correlation coefficient (R^2^ = 0.8) [34]. The measurement of salivary melatonin is advantageous, especially to avoid invasive procedures [35].

### 2.6. Statistical Analysis

The values are expressed as mean ± SEM. The normality of the variables was checked using the Shapiro–Wilk test, followed by Student’s t-test for normally distributed samples or Mann–Whitney test for nonparametric samples. The minimum level of significance was set at *p* < 0.05. Statistical analysis was performed with the SPSS^®^ Statistics v.27.0 package.

## 3. Results

All participants were Caucasian women. FM patients had been diagnosed with FM for more than two years. No significant differences were found between the groups in age and BMI. FM patients showed worse subjective sleep quality (*p* < 0.001) with respect to the RG. The representative groups, RRG and RFM, presented age, BMI, and PSQI score in the same range than all volunteers from RG and FM patient groups. Thus, PSQI was also higher (*p* < 0.001) in the RFM group with respect to RRG (Table 1).

### 3.1. Objective Sleep Quality: Accelerometry Parameters

Table 2 shows actigraphy sleep outcomes. No significant differences were found between the two experimental groups in terms of latency (time it takes a person to fall asleep after turning the lights out), sleep efficiency (ratio of total sleep time and time in bed), WASO (wake after sleep onset), average nightly awakenings, and daily lux average counts (average lux value during scored and non-scored time), measured using accelerometry. Notably, the FM patients went to bed and woke up later than the RG women.

### 3.2. Concentration of Hormones and Neuromediators Related to Stress and Sleep

No significant differences between FM patients and the control group (in accordance with data from our previous study [27]) were found in the systemic levels of oxytocin (1229 ± 171 µg/mL vs. 1248 ± 120 µg/mL, respectively), serotonin (135 ± 14 ng/mL vs. 271 ± 79 ng/mL, respectively), melatonin (13 ± 1 µg/mL vs. 17 ± 2 µg/mL, respectively), adrenaline (31 ± 2 µg/mL vs. 32 ± 4 µg/mL, respectively), and noradrenaline (185 ± 9 µg/mL vs. 168 ± 8 µg/mL, respectively), in this last case with a clear tendency to be higher (*p* < 0.07). Figure 1 represents the adrenaline/noradrenaline ratio (Figure 1A) and serotonin/melatonin ratio (Figure 1B) of their systemic concentrations. As expected, when evaluating the adrenaline/noradrenaline ratio, no significant differences were observed between the groups. However, FM patients (RFM) showed lower (*p* < 0.05) serotonin/melatonin values than those obtained the reference group (RRG).

## 4. Discussion

The relevance of sleep problems in the pathophysiology of FM has generated numerous articles in recent years. Despite this, the causes of this disorder and how it might influence the many symptoms associated with the disease are still being investigated. It is well established that there is a bidirectional neuroimmunoendocrine communication that can affect the different physiological systems, so sleep dysregulation could be exacerbating the alteration of a multitude of disorders such as pain, fatigue, or low mood [36]. In addition, an altered inflammatory state could also be altering the stress response [37] and sleep [38]. Then, it seems very clear that sleep disorders can negatively affect FM symptoms and vice versa [6].

Among the different ways of assessing these sleep problems, subjective measures through questionnaires have great clinical utility due to their simplicity and low cost, but may provide results that do not exactly match those obtained through objective measurements [39]. Very few studies exist in the literature on the objective assessment of sleep in FM patients, particularly through accelerometry. In this context, sleep difficulties in FM appear to be increased when reported subjectively, using sleep quality questionnaires, but tend to have modest correlations when assessed objectively with actigraphic devices [40] or polysomnography [41]. Then, these sleep difficulties may be more a consequence of perception. In this context, our results are consistent with those of other authors who used accelerometry as an objective measure of sleep in women with FM, finding no major differences between them and healthy women [42]. Nevertheless, other authors did find that abnormal nocturnal activity patterns collected using accelerometry were associated with poorer sleep quality and greater FM symptomatology [43,44,45]. This dissonance raises the question of whether psychological disturbances influence the perception of sleep quality, or whether it is really poor sleep quality that contributes to poor mood and prevents emotional recovery from stressful experiences [8,12].

The absence of significant differences with respect to the reference control group in the objective sleep quality of FM patients is also confirmed by the results obtained in the assessment of the main neuroendocrine mediators involved in sleep. For example, the levels of oxytocin, a hormone that is clearly related to sleep quality [46] and also related to pain and depressive states in FM, were similar to those observed in the control group, in agreement with previous research in FM patients not diagnosed with depression [47]. Nevertheless, to the best of our knowledge, there are no studies evaluating the relationship between oxytocin levels and sleep quality in FM patients, which indicates the novelty of the results of the present investigation in this respect.

Regarding catecholamines, results of the present investigation agree with the few studies carried out in this context. It is known that, while adrenaline is related to high levels of stress [46] that can impair sleep, noradrenaline is strongly associated with sleep quality and maintenance [48], and people with sleep disorders have been found to have low plasma noradrenaline levels [46]. The absence of differences in the ratio of adrenaline/noradrenaline concentration seems to be in agreement with the results of accelerometry tests showing that FM patients do not have an objective sleep disturbance, at least related to these neurohormones. In fact, these results are consistent with those from Rus and co-workers [49], who found no differences in adrenaline levels in FM patients compared to healthy people, although they did have increased noradrenaline levels. These results also align with previous results from our research group where higher levels of noradrenaline were observed in FM patients compared with healthy women [19,37]. Nonetheless, the present study is the first to show a link between catecholamines and objective sleep levels in FM patients.

Even more paradoxical were the results obtained for serotonin and melatonin concentrations, which further support the absence of an objectively worse sleep quality in our FM patients, although they seem to be compatible with elevated noradrenaline levels and dysregulation of the stress response [27]. Serotonin and melatonin are two hormones that play a key role in the sleep/wake cycle [50]. In contrast to melatonin, a hormone that signals the body that it is time to sleep when it is dark, serotonin levels tend to peak in the presence of light, and are therefore associated with waking states, while during deep sleep they fall to their minimum. Stress is one of the most common causes of low serotonin levels and this causes a cascade of events: insomnia, depression, daytime fatigue, and anxiety, which in turn causes sleep problems [51]. In fact, there is strong evidence that deficiency in normal serotonergic functioning may be related to the pathophysiology of FM [52]. On the other hand, melatonin has been measured in FM patients in numerous studies, finding different results, ranging from normal [53] to decreased [54] and even increased [55] melatonin levels. It has been suggested that altered melatonin levels may disrupt night-time sleep in FM patients, which could lead to altered pain perception in the early morning [56]. One of the major postulated hypotheses suggests that alterations in melatonin secretion could also cause changes in the daytime secretion of hormones related to the HPA axis [57], and this dysregulation, as mentioned above, could promote specific symptoms of this disease. Furthermore, numerous studies have reported that the potential dysregulation described on the secretion of cortisol, serotonin, cytokines [37,58], and melatonin [54,55] would lead to alterations in the circadian rhythm in these patients, which could contribute to sleep disturbance at night, fatigue during the day, and altered pain perception [56]. Thus, in this context, the serotonin/melatonin ratio is also important in order to evaluate objective sleep quality, and a lower ratio in FM patients compared with the reference group without FM does not seem to be consistent with an impaired sleep quality, reinforcing accelerometry and catecholamine ratio results.

So, the question is: why do FM patients perceive that they sleep poorly? This subjective perception of poor sleep quality could be related to a reduced physiological and psychological capacity to regulate stress responses [19], which could also influence a delay and dysregulation of the rest–activity circadian rhythm that is objectively manifested by delayed bedtime and wake-up times and decreased serotonin/melatonin ratio, more compatible with nocturnal levels in healthy people (according to manufacturing information).

## 5. Conclusions

Considering the results obtained in the present investigation, it cannot be concluded that patients with FM present objectively worse sleep quality than the reference group of women of the same age range without FM. This is corroborated both by accelerometry determinations and by the absence of impaired ratios of the main neuroendocrine biomarkers involved in sleep.

The results of this pilot study in a group of patients diagnosed with FM may contribute to a better understanding of the pathophysiology and aetiology of this syndrome, particularly in relation to sleep and, therefore, to perceived fatigue. The present research also has a number of limitations that will have to be addressed in the future. One of these may be the limited sample size in the second part of the study, related to objective determinations. This will be need to be carried out mainly in clinical tests by increasing the number of patients and assessing the levels of all the evaluated objective biomarkers, but particularly those of melatonin (but also serotonin and catecholamines), over a greater number of hours throughout the day and particularly during night.

## Figures and Tables

**Figure 1 biomedicines-11-01980-f001:**
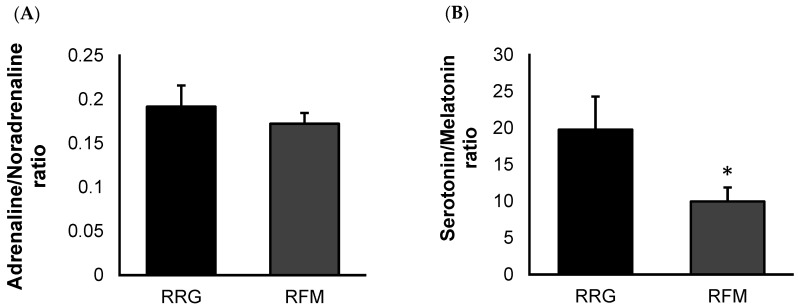
Ratio of systemic concentrations of sleep-related neuromediators from representative FM patient group (RFM; *n* = 11) compared to an age-matched reference group of “healthy” women (RRG; *n* = 11): adrenaline/noradrenaline ratio (**A**), and serotonin/melatonin ratio (**B**). Values are expressed by the mean ± SEM in each group. * *p* < 0.05 with respect to RRG.

**Table 1 biomedicines-11-01980-t001:** Anthropometric characteristics and subjective sleep quality of the entire and representative groups.

	RG (*n* = 68)	FM Patients (*n* = 68)	RRG (*n* = 11)	RFM (*n* = 11)
Gender (%)	Women (100%)	Women (100%)	Women (100%)	Women (100%)
Ethnic group (%)	Caucasian (100%)	Caucasian (100%)	Caucasian (100%)	Caucasian (100%)
Age (years)	54.93 ± 8.22	55.87 ± 7.25	55.81 ± 2.08	57.91 ± 4.18
BMI (kg/m^2^)	25.61 ± 4.76	26.56 ± 5.27	24.62 ± 0.83	27.30 ± 3.79
Duration of FM diagnosed (years)	-	>2	-	>2
PSQI score	5.86 ± 0.47	14.47 ± 0.44 ***	6.27 ± 1.03	15.73 ± 0.79 ***

Data are expressed as mean ± SEM and as percentage (%). RG: reference group; FM: fibromyalgia; BMI: body mass index; PSQI: Pittsburgh Sleep Quality Index; RRG: representative reference group; RFM: representative FM patient group. *** *p* < 0.001 with respect to each corresponding reference group.

**Table 2 biomedicines-11-01980-t002:** Objective sleep-related parameters measured by accelerometry.

	RRG (*n* = 11)	RFM (*n* = 11)
Retire to bed (hour:min)	23:11	0:09 **
Rise from bed (hour:min)	6:51	8:10 **
Latency (min)	1.11 ± 0.37	0.71 ± 0.15
Efficiency (%)	88.70 ± 1.26	89.14 ± 1.01
WASO (min)	48.08 ± 4.89	48.57 ± 5.03
N° of awakenings	13.61 ± 1.65	11.82 ± 1.36
Daily lux average count (<1 min)	57.81 ± 10.91	55.85 ± 15.29

Data are expressed as mean ± SEM. PSQI: Pittsburgh Sleep Quality Index; WASO: wake after sleep onset; RRG: representative reference group; RFM: representative FM patient group. ** *p* < 0.01.

## Data Availability

The raw data supporting the conclusions of the manuscript will be made available by the authors, without undue reservation, to any qualified researcher.

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
