# Peer review of "Subjective Sleep Quality Versus Objective Accelerometric Measures of Sleep and Systemic Concentrations of Sleep-Related Hormones as Objective Biomarkers in Fibromyalgia Patients"

_biomedicines, 2023, doi:10.3390/biomedicines11071980_

Round 1

Reviewer 1 Report

Thankyou for giving me the possibility to review the paper "Subjective Sleep Quality Versus Objective Accelerometric Measures of Sleep and Systemic Concentrations of Sleep-Re)ated Hormones as Objective Biomarkers in Fibromyalgia Patients". The paper deals with a very interesting topic, since it aims to determine if the poor perceived sleep quality in FM patients could be corroborated by objective physiological determinations.

However, before considering it for pubblication, the following concerns should be addressed:

1. Discussion should be summarized and improved, emphasizing better the findings of yhe present study 

2. Add conclusions

3. Limit the number of self-citations to two.

Minor English revision 

Author Response

Reviewer 1.

Thank you for giving me the possibility to review the paper "Subjective Sleep Quality Versus Objective Accelerometric Measures of Sleep and Systemic Concentrations of Sleep-Re)ated Hormones as Objective Biomarkers in Fibromyalgia Patients". The paper deals with a very interesting topic, since it aims to determine if the poor perceived sleep quality in FM patients could be corroborated by objective physiological determinations.

However, before considering it for pubblication, the following concerns should be addressed:

  1. Discussion should be summarized and improved, emphasizing better the findings of the present study.
  2. Add conclusions

First of all, we would like to thank you for your very positive comments and constructive criticisms.

We have abridged and improved the discussion, also including a separate paragraph of conclusions (as suggested) and clarifying the potential limitation of the study (as suggested by another referee) (in red). We also feel that the discussion is now really improved.

  1. Limit the number of self-citations to two

We think that 6 self-citations (of a total of 59) is not too much, taking into account the need to discuss results (without other possibilities and also taking into account the manuscript instructions "Authors should not engage in excessive self-citation of their own work". Nevertheless, we have eliminated two of these.

Reviewer 2 Report

This study follows a previous investigation from these authors and addresses the use of accelerometry tests and ratios of systemic concentratins of hormones and neuromediators related to sleep quality in fibromyalgia patients, with a particular focus on adrenaline/noradrenaline and serotonin/melatonin ratios. 

Line 63: You mention your previous study, please provide the reference even if this study is in print.

Line 74: Change "68" to "Sixty eight"

Line 82: Change "Inventory" to "Index"

Line 114: Change "Pittsburgh Sleep Index Quality" to "Pittsburgh Sleep Quality Index"

In Table 2 I suggest changing "In bed" and "Out bed" to "Retire" and "Rise" or "Retire to bed" and "Rise from bed"

Line 175: Change "according with data..." to "in accordance with data..."

The Discussion is clearly written and does not overstate the findings.

Generally the manuscript is well written and clear. I made a few minor suggestions in my comments above.

Author Response

Referee 2.

This study follows a previous investigation from these authors and addresses the use of accelerometry tests and ratios of systemic concentratins of hormones and neuromediators related to sleep quality in fibromyalgia patients, with a particular focus on adrenaline/noradrenaline and serotonin/melatonin ratios. 

Line 63: You mention your previous study, please provide the reference even if this study is in print.

Reference provided (in red), thank you.

Line 74: Change "68" to "Sixty eight"

We have changed it (in red).

Line 82: Change "Inventory" to "Index"

We have changed it (in red).

Line 114: Change "Pittsburgh Sleep Index Quality" to "Pittsburgh Sleep Quality Index"

We have corrected it (in red).

In Table 2 I suggest changing "In bed" and "Out bed" to "Retire" and "Rise" or "Retire to bed" and "Rise from bed"

Thank you for your suggestion. We have changed it (in red).

Line 175: Change "according with data..." to "in accordance with data..."

We have changed it (in red).

The Discussion is clearly written and does not overstate the findings.

Thank you for your positive comments.

Reviewer 3 Report

The Authors aimed to evaluate a correlation between sleep quality and fibromyalgia symptoms.

The topic is interesting and the study well designed.

Introduction ok

Methods: how was sample size calculated?

Was assumption of any drug which might influence sleep activity and exclusion criterium?

Why was sample size of "part two" different?please detail and acknowledge it as a limitation.

Table 1. Please add statistical comparison between groups. Were they homogeneous?

table 2 should be detailed further.

Discussion. Do the Authors believe that sleep disorder can negative affect FM symptoms or viceversa?Please discuss.

Commas should be changed in dots in tables.

Minor spelling errors

Author Response

Referee 3.

The Authors aimed to evaluate a correlation between sleep quality and fibromyalgia symptoms.

The topic is interesting and the study well designed.

Introduction ok

Methods: how was sample size calculated?

As indicated in the Methods section (first paragraph), n=68 women was the sample size of ALL FM volunteers from the FM association of Extremadura (Spain) who met the inclusion criteria, as a very homogeneous group. Then, the corresponding control group was also selected.

Was assumption of any drug which might influence sleep activity and exclusion criterium?

Yes. As indicated in line 106, prescription drugs related to improving sleep quality were restricted.

Why was sample size of "part two" different?please detail and acknowledge it as a limitation.

It is explained in the second part of the Methods section. According to your suggestions, we have indicated it as a potential limitation of the study in the discussion (in red).

Table 1. Please add statistical comparison between groups. Were they homogeneous?

Statistical comparison between groups was added, but only significant differences were indicated by the corresponding asterisk and p-value. Yes, you are right, they were very homogeneous groups.

table 2 should be detailed further.

As also suggested by other referees, we have changed “in bed” to “retire to bed” and “out bed” to “rise from bed”. In addition, the meaning of the sleep parameters is also explained (in red).

Discussion. Do the Authors believe that sleep disorder can negative affect FM symptoms or vice versa? Please discuss.

Yes. In fact, it is already indicated in both the second paragraph of the introduction and the first paragraph of the discussion. Nevertheless, following your suggestions, we have reinforced the idea by including at the end of the first paragraph of the discussion: “Then, it seems very clear that sleep disorders can negatively affect FM symptoms and vice versa [6] “ (in red)

Commas should be changed in dots in tables

We have changed.

Round 2

Reviewer 1 Report

Thankyou for giving me the possibility to review the revised version of the manuscript  

It is now suitable for publication in the present form. 

Accept in present form